# Impact of sharing electronic health records with patients on the quality and safety of care: a systematic review and narrative synthesis protocol

Ana Luisa Neves,[1,2] Alexander W Carter,[1] Lisa Freise,[1] Liliana Laranjo,[3] Ara Darzi,[1] Erik K Mayer[1]

[1]Center for Health Policy, Institute of Global Health Innovation, Imperial College London, London, UK
[2]CINTESIS – Center for Health Technology and Services Research, University of Porto, Porto, Portugal
[3]Australian Institute of Health Innovation, Centre for Health Informatics, Macquarie University, Macquarie Park, New South Wales, Australia

**Correspondence to**
Dr Ana Luisa Neves;
ana.luisa.neves14@ic.ac.uk

## ABSTRACT

**Introduction** Providing patients with access to electronic health records (EHRs) has emerged as a promising solution to improve quality of care and safety. As the efforts to develop and implement EHR-based data sharing platforms mature and scale up worldwide, there is a need to evaluate the impact of these interventions and to weigh their relative risks and benefits, in order to inform evidence-based health policies. The aim of this work is to systematically characterise and appraise the demonstrated benefits and risks of sharing EHR with patients, by mapping them across the six domains of quality of care of the Institute of Medicine (IOM) analytical framework (ie, patient-centredness, effectiveness, efficiency, timeliness, equity and safety).

**Methods and analysis** CINAHL, Cochrane, Embase, HMIC, Medline/PubMed and PsycINFO databases will be searched from January 1997 to August 2017. Primary outcomes will include measures related with the six domains of quality of care of the IOM analytical framework. The quality of the studies will be assessed using the Cochrane Risk of Bias Tool, the ROBINS-I Tool and the Drummond's checklist. A narrative synthesis will be conducted for all included studies. Subgroup analysis will be performed by domain of quality of care domain and by time scale (ie, short-term, medium-term or long-term impact). The body of evidence will be summarised in a Summary of Findings table and its strength assessed according to the GRADE criteria.

**Ethics and dissemination** This review does not require ethical approval as it will summarise published studies with non-identifiable data. This protocol complies with the Preferred Reporting Items for Systematic Review and Meta-Analyses Protocols guidelines. Findings will be disseminated widely through peer-reviewed publication and conference presentations, and patient partners will be included in summarising the research findings into lay summaries and reports.

**PROSPERO registration number** CRD42017070092.

## Strengths and limitations of this study

► Comprehensive characterisation of interventions sharing electronic health records with patients.
► Summary and appraisal of existing evidence on the potential benefits and risks of these interventions, grouped by domain of quality of care.
► Map the contribution of these interventions in short-term, medium-term and long-term time frames, in order to customise informed decisions in health policies.
► Expected limitations include the heterogeneous nature of the outcomes assessed and the potentially reduced sample size in subgroup analyses, which may negatively impact the statistical power in data synthesis.

controllers, through a cumbersome procedural process.[1] Additionally, as health information is fragmented between different organisations and levels of care, data access requests are often unable to provide a comprehensive health history record.[2 3]

In the last decade, electronic health records (EHR) have emerged as a promising solution to enhance patients' access to centralised medical information.[4] The adoption of EHR by primary care practices, hospitals and other healthcare organisations has steadily increased in the last years. In England, the percentage of general practice surgeries that allowed patients to access their medical records online increased from 3% to 97% between April 2014 and February 2016.[5] Patients' willingness and ability to access their health information through web portals is influenced by both individual (ie, age, ethnicity, education level, health literacy and health status) and by healthcare delivery factors (ie, provider endorsement and portal usability).[6 7] Various EHR-based platforms are currently used to share health information

## INTRODUCTION

Although, in England, patients have had the legal right to access their health records since 1998, access to paper-based health records is mediated by health professionals and data

with patients, including direct online access, with or without patient–provider communication systems[8 9], and health maintenance reminders.[10 11] As these efforts mature and scale up worldwide, there is a need to evaluate the impact of interventions sharing EHR with patients, in order to weigh their relative risks and benefits and inform evidence-based health policies.

The Institute of Medicine (IOM) identified six domains of healthcare quality: patient-centredness, effectiveness, efficiency, timeliness, equity and safety.[12] Patient-centredness ensures that the care provided respects and responds to individual patient preferences, needs and values, thus incorporating these in clinical decisions.[12 13] Healthcare shall provide evidence-based services, which can be ultimately expressed as improvements in health outcomes (ie, effectiveness),[14] while ensuring patient safety (ie, prevention of errors and adverse effects associated with healthcare).[12] Other aspects of quality of delivery of care include the minimisation of the waste of human, physical or economical resources (ie, efficiency), the reduction of waits and harmful delays (ie, timeliness), and the reduction of avoidable differences on the delivery of care between different groups of healthcare users (ie, equity).[12 15 16]

Providing patients with access to their health records has been linked to theorised benefits in four major domains of healthcare quality: patient-centredness, effectiveness, safety and efficiency.[17–19] However, despite the growing body of evidence on the theorised benefits of sharing EHR with patients on these domains, there is still a considerable gap between the predicted and demonstrated benefits of these interventions.[20]

In order to analyse the effect of providing patients access to their medical records on quality outcomes, Davis Giardina et al performed a systematic review including studies published between 1970 and 2012.[21] According to this work, a limited amount of evidence suggests that access to medical records improves patient satisfaction and enhances patient–provider communication.[21] Similarly, a systematic review from de Lusignan et al reported that providing patients online access to their EHR increased convenience and satisfaction.[22] These findings are in line with the model proposed by Otte-Trojel et al, according to which sharing EHR with patients can improve both patient–provider communication and patient satisfaction, by increasing continuity of care and patient convenience, respectively.[23]

Conversely, no clear benefits were found on effectiveness.[21] Until 2012, only a few studies evaluated the impact on effectiveness, most focusing on type 2 diabetes, and with inconsistent results. Tenforde et al showed that providing access to medical records was associated with lower glycated haemoglobin A1c values[24]; however, no significant effect was found in other studies assessing diabetes-specific effectiveness measures.[25 26] One of the limitations of this review consists in the inclusion of studies evaluating the impact of sharing both electronic and paper-based health records—and this heterogeneity might mask potential specific benefits and risks of sharing EHR with patients. Furthermore, as pointed out by the authors, the paucity of papers published up to that date resulted in a tendency to include small and methodologically less robust studies, thus increasing the risk of selective reporting and/or publication bias.[21] Mold et al also performed a systematic review assessing the impact of providing patients with access to their EHR; based in studies published between 1999 and 2012, this work found a positive influence in patient safety.[27]

However, the authors were unable to find a consistent beneficial effect on efficiency measures (ie, number of face-to-face visits and telephone appointments) in both reviews.[21 27] While some studies reported an increase in the number of face-to-face consultations,[8 28] others document a decrease.[11 29 30] Similarly, inconsistent results were found regarding the impact on telephone consultations: only one study reported a decline in the total number,[31] while six other studies reported either no change or an increase.[9 28–30 32 33] It is important to note, however, that most of the included studies assessing efficiency measures included in this review showed a high risk of bias, mostly related to either unclear or absent blinding methods.[27]

The landmark reviews of Mold et al,[27] Davis Giardina et al,[21] Ammenwerth et al [34], and Goldzweig et al[7] provide a comprehensive characterisation of the literature published until 2013, highlighting the paucity and the scientific limitations of the evidence published until that date. Although these reviews were unable to demonstrate clear benefits on efficiency and effectiveness measures, the debates around patients' rights and data ownership in the digital era, and the need to improve patient-centredness of healthcare delivery have acted as strong drivers to allocate resources to interventions and platforms aiming to share EHR with patients. As consequence of these efforts, it is plausible that studies performed in the last 5 years can provide further clarification for this evidence gap.

Furthermore, systematic reviews performed to date do not address all domains of quality of care; in particular, the impact of sharing EHR with patients on timeliness or equity has not been addressed.[7 21 27 34 35] This is a particularly relevant gap in knowledge, given that interventions aimed at improving the quality of care do not necessarily improve all specific domains, and may even have a deleterious effect in some of them.

This review will expand on the above-mentioned work, in order to identify recent methodological and scientific progress until June 2017. Following the Preferred Reporting Items for Systematic Reviews and Meta-Analyses Protocols (PRISMA-P) checklist as guidance,[36] we propose a systematic and reproducible strategy to query the literature on the demonstrated benefits and risks of sharing EHR with patients, and map these results in a comprehensive framework of healthcare quality measures.

## Which are the demonstrated **benefits and risks** of sharing EHR with patients?

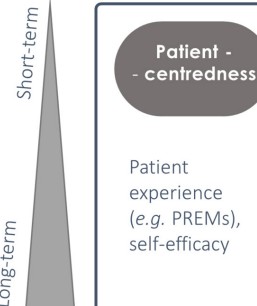
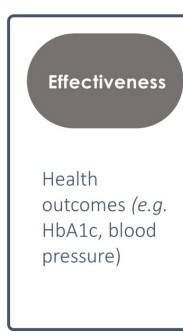
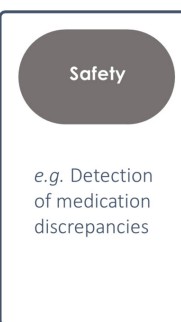
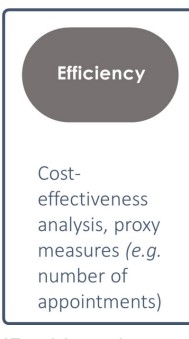
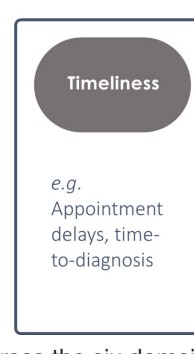
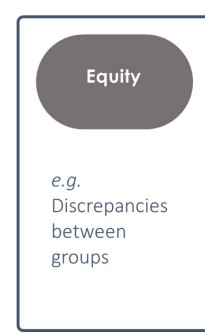

Short-term

Long-term

| Patient-centredness | Effectiveness | Safety | Efficiency | Timeliness | Equity |
|---|---|---|---|---|---|
| Patient experience (*e.g.* PREMs), self-efficacy | Health outcomes (*e.g.* HbA1c, blood pressure) | *e.g.* Detection of medication discrepancies | Cost-effectiveness analysis, proxy measures (*e.g.* number of appointments) | *e.g.* Appointment delays, time-to-diagnosis | *e.g.* Discrepancies between groups |

**Figure 1**  Mapping the demonstrated benefits and risks of sharing EHR with patients across the six domains of quality of care, as previously defined by the Institute of Medicine analytical framework.[12] Subgroup analysis will be performed by domain of quality of care and by time scale. EHR, electronic health records; HbA1c, haemoglobin A1c; PREMs, patient-reported experience measures.

## RESEARCH AIMS

The main objectives of this review are: (1) to systematically characterise interventions sharing EHR with patients, and (2) to assess the demonstrated risks and benefits of these interventions on patient-centredness, effectiveness, safety, efficiency, timeliness and equity, compared with usual care (no intervention). As secondary aim, we will map the contribution of these interventions in short-term, medium-term and long-term time frames (figure 1).

## METHODS AND ANALYSIS
### Search strategy

The search strategy will be performed using resources that enhance methodological transparency and improve the reproducibility of the results and evidence synthesis. A search of the literature from the last 20 years (January 1997 to August 2017) will be performed on CINAHL, Cochrane, Embase, HMIC, Medline/PubMed and PsycINFO. Search strings (table 1) will combine free terms and controlled vocabulary, whenever supported. We will also search grey literature sources, including registrations in the International Prospective Register of Systematic Reviews, reports of relevant stakeholder organisations (National Health Service Digital, American Medical Informatics Association (AMIA), eHealth at WHO, International Society for Telemedicine and eHealth), and conference proceedings (last 5 years) of several related conferences (AMIA, MedInfo, Medicine 2.0, Medicine X), in order to identify possible additional studies that meet the inclusion criteria. Language restrictions will be applied and only articles in English will be included.

### Study selection criteria

A summary of the participants, interventions, comparators and outcomes considered, as well as the type of studies included, is provided in table 2.

The systematic review will focus on studies on adult subjects, including both patients and carers (mean age of study sample ≥16 years). The systematic review does not focus on a particular disease area or health system setting as it intends to comprehensively characterise the scope of interventions sharing EHR with patients.

Studies assessing the impact of sharing EHR with patients, either isolated or as part of a multicomponent intervention, will be included. Included interventions will comprise: (1) web-based patient access to EHR; (2) EHR-based health reminders/messaging or (3) online patient–provider communication systems (health information exchange platforms). Studies focusing on health reminders only (not EHR based) or appointment reminders will not be considered. The comparator will be 'no intervention' (eg, usual care).

Primary outcomes will include any measure related to (1) patient-centredness (eg, patient-reported experience measures), (2) effectiveness (eg, health outcomes); (3) patient safety (eg, identification of medication discrepancies); (4) efficiency (eg, economic evaluation measures and proxies, including service costs, number of consultations/admissions), (5) timeliness (eg, waiting lists, time to treatment) and (6) equity (eg, discrepancies in quality measures between different groups of patients) (figure 1). Studies that only report cognitive outcomes (eg, intent), motivational outcomes or other subjective psychological measures will be excluded. The types of study considered in this systematic review will be (1) randomised controlled trials; (2) cluster randomised trials; (3) quasi-experimental studies; (4) case–control studies, (5) cohort studies and (6) cost-effectiveness studies. The reference lists of systematic reviews identified in this search will also be screened to ensure all eligible studies are captured.

**Table 1** Concepts and search items

| Database | Search items |
|---|---|
| CINAHL via EBSCO | 1. (((electronic* or online or on-line or digital*) N1 (health record* or medical record* or personal record* or patient record*)) or EHR# or EMR# or ephr#)<br>2. ((information or data) N4 (shar* or exchang*)) or HIE or HIEs or access*)<br>3. #1 and #2<br>4. (((experience or satisfaction) N4 (patient* or consumer* or client* or survey or questionnaire*)) or PREM* or patient-reported experience measure*)<br>5.(effectiveness or health outcome*)<br>6. (patient N1 (safety or harm)) or safety manag* or accident prevent* or error* or medication reconcil* or near miss*<br>7. (efficiency or economic* or cost* or expenditure* or charge* or fee*1 or (number N1 appointment*) or (number adj2 admission*) or (number N1 telephone visit*))<br>8. waiting lists or timeliness or time-to-treatment<br>9. health equity<br>10. #4 or #5 or #6 or #7 or #8 or #9<br>13. 3 and 10 |
| Cochrane via url: http://onlinelibrary.wiley.com/ cochranelibrary/search/advanced | 1. (((electronic* or online or on-line or digital*) near/1 (health record* or medical record* or personal record* or patient record*)) or EHR or EHRs or EMR or EMRs or ephr or ephrs)<br>2. Electronic Health Records [MesH]<br>3. #1 or #2<br>4. (((information or data) near/4 (shar* or exchang*)) or HIE or HIES or access*)<br>5. Information Dissemination [MesH]<br>6. #4 or #5<br>7. #3 and #6<br>8. (((experience or satisfaction) near/4 (patient* or consumer* or client* or survey or questionnaire*)) or PREM* or patient-reported experience measure*)<br>9. (effectiveness or health outcome*)<br>10. (patient near/1 safety) or (patient near/1 harm) or safety manag* or accident prevent* or error* or medication reconcil*<br>11. (efficiency or economic* or cost* or expenditure* or charge* or fee* or (number near/1 appointment*) or (number near/1 admission*) or (number near/1 telephone visit*))<br>12. time-to-treatment or timeliness<br>13. waiting lists [MesH]<br>14. health equity [MesH]<br>15. #8 or #9 or #10 or #11 or #12 or #13 or #14<br>17. 7 and 15 |
| Embase via Ovid | 1. (((electronic* or online or on-line or digital*) adj2 (health record* or medical record* or personal record* or patient record*)) or EHR? or EMR? or ephr?)<br>2. Electronic health record/<br>3. 1 or 2<br>4. (((information or data) adj5 (shar* or exchang*)) or HIE*2 or access*)<br>5. Information dissemination/<br>6. 4 or 5<br>7. 3 and 6<br>8. (((experience or satisfaction) adj5 (patient* or consumer* or client* or survey or questionnaire*)) or PREM* or patient-reported experience measure*)<br>9. (effectiveness or health outcome*) 10. (patient adj2 (safety or harm)) or safety manag* or accident prevent* or error* or medication reconcil* or near miss*<br>11. (efficiency or economic* or cost* or expenditure* or charge* or fee*1 or (number adj2 appointment*) or (number adj2 admission*) or (number adj2 telephone visit*))<br>12. waiting list* or time to treatment/or timeliness<br>13. health equity/<br>14. 8 or 9 or 10 11 or 12 or 13<br>15. 7 and 14 |
| HMIC via Ovid | 1. (((electronic* or online or on-line or digital*) adj2 (health record* or medical record* or personal record* or patient record*)) or EHR? or EMR? or ephr?)<br>2. Electronic patient records/<br>3. 1 or 2<br>4. (((information or data) adj5 (shar* or exchang*)) or HIE*2 or access*)<br>5. Information exchange/<br>6. 4 or 5<br>7. 3 and 6<br>8. (((experience or satisfaction) adj5 (patient* or consumer* or client* or survey or questionnaire*)) or PREM* or patient-reported experience measure*)<br>9. (effectiveness or health outcome*) 10. (patient adj2 (safety or harm)) or safety manag* or accident prevent* or error* or medication reconcil* or near miss*<br>11. (efficiency or economic* or cost* or expenditure* or charge* or fee*1 or (number adj2 appointment*) or (number adj2 admission*) or (number adj2 telephone visit*))<br>12. waiting lists/or patient waiting time or timeliness<br>13. health inequalities/or equity<br>14. 8 or 9 or 10 or 11 or 12 or 13<br>15. 7 and 14 |

**Table 1** Continued

| Database | Search items |
|---|---|
| Medline via Ovid | 1. (((electronic* or online or on-line or digital*) adj2 (health record* or medical record* or personal record* or patient record*)) or EHR? or EMR? or ephr?)<br>2. Electronic Health Records/<br>3. 1 or 2<br>4. (((information or data) adj5 (shar* or exchang*)) or HIE*2 or access*<br>5. Information Dissemination/<br>6. 4 or 5<br>7. 3 and 6<br>8. (((experience or satisfaction) adj5 (patient* or consumer* or client* or survey or questionnaire*)) or PREM* or patient-reported experience measure*)<br>9. (effectiveness or health outcome*)<br>10. ((patient adj2 (safety or harm)) or safety manag* or accident prevent* or error* or medication reconcil* or near miss*)<br>11. (efficiency or economic* or cost* or expenditure* or charge* or fee*1 or (number adj2 appointment*) or (number adj2 admission*) or (number adj2 telephone visit*))<br>12. Waiting Lists/or Time-to-treatment/or timeliness<br>13. Health Equity/<br>14. 8 or 9 or 10 or 11 or 12 or 13 15. 7 and 14 |
| PsycINFO via Ovid | 1. (((electronic* or online or on-line or digital*) adj2 (health record* or medical record* or personal record* or patient record*)) or EHR? or EMR? or ephr?)<br>2. (((information or data) adj5 (shar* or exchang*)) or HIE*2 or access*)<br>3. 1 and 2<br>4. (((experience or satisfaction) adj5 (patient* or consumer* or client* or survey or questionnaire*)) or PREM* or patient-reported experience measure*)<br>5. (effectiveness or health outcome*)<br>6. (patient adj2 (safety or harm)) or safety manag* or accident prevent* or error* or medication reconcil* or near miss*<br>7. (efficiency or economic* or cost* or expenditure* or charge* or fee*1 or (number adj2 appointment*) or (number adj2 admission*) or (number adj2 telephone visit*))<br>8. waiting list* or time-to-treatment or timeliness<br>9. equity or health disparities/<br>10. 4 or 5 or 6 or 7 or 8 or 9<br>11. 3 and 10 |

Search themes (facets) and terms derived for each theme relating to the use of EHR and predicted benefits (patient experience, effectiveness and efficiency)

## Screening and data extraction

Quantitative studies will be independently assessed by three reviewers and reported using the PRISMA-P flow diagram.[36] Initial screening of studies will be based on the information contained in their titles and abstracts and will be conducted by two independent investigators. Full-paper screening will be conducted by the same independent investigators. Cohen's kappa will be used to measure intercoder agreement in each screening phase. When there are doubts regarding inclusion or exclusion, a third investigator will be involved in the decision. Two independent investigators will extract information from

**Table 2** Inclusion and exclusion criteria

| | Inclusion criteria | Exclusion criteria |
|---|---|---|
| Population | Adult subjects (patients and carers). | Individuals 16 years of age and under (eg, mean age of study sample <16). |
| Intervention | EHR-based interventions, including:<br>► Patient access to EHR.<br>► EHR-based reminders/messaging.<br>► Unidirectional or bidirectional online patient–provider communication systems (care information exchange platforms). | Health reminders only. |
| Comparison | No intervention (eg, usual care) | |
| Outcome | Any measure related to (1) patient-centredness (eg, patient-reported experience measures), (2) effectiveness (eg, health outcomes); (3) patient safety (eg, identification of medication discrepancies); (4) efficiency (eg, economic evaluation measures and proxies, including service costs, no of consultations/admissions), (5) timeliness (eg, waiting lists, time to treatment) or (5) equity (eg, discrepancies in quality measures between different groups of patients). | Studies that only report cognitive outcomes (eg, intention to), motivational outcomes or other subjective psychological measures. |
| Study type | Randomised controlled trials, cluster randomised trials, quasi-experimental, case–control, cohort studies, cost-effectiveness. | |

the included studies into a standardised form. The data collected for each study will include: name of the first author, year of publication, technology, intervention components and characteristics, study duration, participants' and setting characteristics, outcomes and retention rates. Two investigators will review the abstraction form for consistency. Disagreements will be resolved by a third investigator.

## Quality assessment

The quality of randomised controlled trials and cluster randomised trials will be assessed using the Cochrane Risk of Bias Tool,[37] that assesses the following study-level aspects: (1) randomisation sequence allocation; (2) allocation concealment; (3) blinding; (4) completeness of outcome data and (4) selective outcome reporting. The quality of non-randomised intervention studies (ie, case control, cohort, quasi-experimental) will be appraised using the 'Risk of Bias In Non-Randomised Studies - of Interventions' (ROBINS-I) tool, which assesses bias due to (1) confounding, (2) selection of participants, (3) classification of interventions, (4) deviations from intended interventions, (5) missing data, (6) measurement of outcomes and (7) selection of reported results.[38] For cost-effectiveness studies, the Drummond's checklist for assessing economic evaluations will be used.[39] Two independent reviewers will score the selected studies and disagreements will be resolved by a third person.

The risk of bias for each outcome across individual studies will be summarised as a narrative statement, and supported by a risk of bias table. A review-level narrative summary of the risk of bias will also be provided.

## Descriptive analysis and meta-analysis

Planned subgroup analysis will be performed by domain of quality of care (IOM framework) and by time scale (ie, short-term, medium-term or long-term impact). For studies with a high or unclear risk of bias, defined as high or unclear risk in 50% or more of the quality assessment outcomes, a narrative description of the risk of bias will be provided. Risk of bias assessments will be incorporated into synthesis by performing sensitivity analysis (ie, limiting to studies at lowest risk of bias in a secondary analysis). Depending on the amount of information retrieved, subgroup analysis will also be performed for specific diseases.

A narrative synthesis will be conducted for all the included studies. Parallel-group trials that are deemed comparable in relevant ways will be pooled together for a summary effect. Whenever possible, continuous and dichotomous outcomes will be pooled together for meta-analysis purposes. All effect sizes will be transformed into a common metric, in order to make them comparable across studies—the bias-corrected standardised difference in means (Hedges' g)—classified as positive when in favour of the intervention and negative when in favour of the control. Heterogeneity will be assessed using $I^2$. The presence of publication bias will be evaluated by use of a funnel plot and the Duval and Tweedie's trim and fill method.[40]

The body of evidence will be summarised in a Summary of Findings table and the strength of the body of evidence will be assessed according to the 'Grading of Recommendations Assessment, Development and Evaluation' (GRADE) criteria.[41]

## Patient and public involvement

Our research question emerged from the implementation evaluation of the care information exchange (CIE), a pilot web portal/patient-controlled EHR happening across a 2.4 million population in Northwest London. CIE implementation evaluation was shaped by its steering group, which included lay partners, and their perspectives reinforced that our research question was relevant and aligned with patients' interest.

Patients were not directly involved in the design of this study. As this is a protocol for a systematic review and no participant recruitment will take place, their involvement on the recruitment and dissemination of findings to participants was not applicable.

However, patient partners will be included in the interpretation of our results, in the co-development of a dissemination strategy, and in summarising the research findings into lay summaries and reports, in order to raise awareness and stimulate public participation on this topic.

## Amendments

Any amendments to this protocol will be documented with reference to saved searches and analysis methods, which will be recorded in bibliographic databases (Ovid), Endnote and Excel templates for data collection and synthesis.

## DISCUSSION

As the implementation of interventions to share EHR with patients scales up worldwide, the systematic evaluation of their impact emerges as a priority research topic.

One of the strengths of the proposed study is to apply a reproducible and transparent procedure for systematic review of the literature. In this protocol, we clearly describe the types of studies, participants, interventions and outcomes that will be included, as well as the data sources, search strategy, data extraction methods (including quality assessment) and methods of combining data.[42] By publishing the research protocol, we reinforce the clarity of the strategy and minimise the risk of bias, namely selective outcome reporting.[37] Second, we will focus solely on the impact of EHR-based studies, increasing the sensitivity to detect specific benefits of this type of intervention. Third, for the first time, we aim to comprehensively evaluate both the benefits and risks of these interventions in a wide range of domains of quality of care, as defined by the IOM, and in diverse time frames. This results shall provide high-level information to inform, support and customise decisions in health policies.

Potential limitations of this study include the heterogeneity of measures and outcomes evaluated and the potentially reduced number of studies in subgroup analyses,

which may negatively influence the statistical power in data synthesis.

**Acknowledgements** We thank Jacqueline Cousins (Library Manager and Liaison Librarian at Imperial College London) for the support improving the composition of the search terms and procedural aspects of the search strategy. We would also like to acknowledge Dr Søren Rud Kristensen (Senior Lecturer in Health Economics, Centre for Health Policy, Institute of Global Health Innovation) and Dr Joachim Marti (Honorary Senior Lecturer in Health Economics, Centre for Health Policy, Institute of Global Health Innovation) for their advice regarding the quality appraisal of cost-effectiveness studies, and Anna-Lawrence Jones (Patient and Public Involvement and Engagement Manager at Imperial National Institute for Health Research (NIHR) Patient Safety Translation Research Centre) for the useful discussions around public and patient involvement.

**Contributors** ALN, AWC and EM conceptualised this research. ALN, AWC and LL designed the protocol. ALN and LF defined the concepts and search items. EM and AD contributed to the conceptualisation and commented on the multiple versions of the protocol. The manuscript was written by ALN with contributions from all authors.

**Funding** The research was supported by the Sowerby Foundation and by the National Institute for Health Research (NIHR) Imperial Patient Safety Translation Research Centre. Infrastructure support was provided by the NIHR Imperial Biomedical Research Centre (BRC).

**Disclaimer** The views of the authors do not necessarily reflect those of the NHS, NIHR or the Department of Health.

**Competing interests** None declared.

**Patient consent** Not required.

**Provenance and peer review** Not commissioned; externally peer reviewed.

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
