## [Reviewer comments · BMJ Open]

ARTICLE DETAILS

TITLE (PROVISIONAL)	Impact of sharing electronic health records with patients on the quality and safety of care: a systematic review and narrative synthesis protocol
AUTHORS	Neves, Ana Luisa; Carter, Alexander; Freise, Lisa; Laranjo, Liliana; Darzi, Ara; Mayer, Erik

VERSION 1 – REVIEW

REVIEWER	Albert Farre University of Birmingham, UK
REVIEW RETURNED	22-Dec-2017

GENERAL COMMENTS	Thank you for providing me with the opportunity to review this manuscript, which is clearly written and well structured, addressing important and timely issues relating to patients' access to electronic health records. I suggest a few possible areas of improvement that the authors might want to take into consideration: Overarching considerations: In my opinion, attention should be paid to accuracy and consistency in the use of terminology in relation to both their approach to synthesis and the intervention under analysis, to ensure that the proposed work is well understood: • Approach to synthesis: The paper is initially framed as a mapping exercise informed by a particular framework (IOM's domains of quality of care) and as a narrative synthesis, and there is also the aim to develop a conceptual model. These would suggest that a thematic (i.e. narrative) or framework based synthesis method would be appropriate; however, in the methods section (lines 215-229) the authors outline an analysis plan relying on methods that produce descriptive and/or inferential statistics, which would imply a different approach to synthesis. This should be clarified and efforts made to ensure consistency throughout the protocol, particularly in relation to the review aims and questions.• Intervention under analysis: The review is described as exploring 'the impact of electronic data sharing on quality of care and safety' and the interventions of interest are referred to as 'EHR-based data sharing' interventions throughout. However, based on both the introduction provided and my understanding of the proposed protocol, the review is specifically concerned with interventions relating to patients' access to EHR. In my opinion this should be clarified and accurate terminology used, given that the term 'data
---

sharing' in the context of healthcare information technologies could relate to a vast range of issues and interventions (e.g. interfacing issues between different systems within organisations, access to patient data across organisations and levels of care, etc.).

Tied to the latter, if the review is specifically concerned with interventions relating to patients' access to HER, it is paramount that the authors' address in the introduction how their proposed review differs from the following Chocrane Review protocol:

- Ammenwerth E, Lannig S, Hörbst A, Muller G, Schnell-Inderst P. Adult patient access to electronic health records (Protocol). Cochrane Database of Systematic Reviews 2017, Issue 6. Art. No.: CD012707. DOI: 10.1002/14651858.CD012707.

Similarly, further clarification could be provided in the introduction as to what this review adds in relation to other relevant existing reviews (in addition to refs 17-20 and 24) which seem to have been missed:

- Ammenwerth E, Schnell-Inderst P, Hoerbst A. The impact of electronic patient portals on patient care: a systematic review of controlled trials. *Journal of Medical Internet Research* 2012;14(6):e162.
- Goldzweig CL, Orshansky G, Paige NM, Towfigh AA, Haggstrom DA, Miake-Lye I, et al. Electronic patient portals: evidence on health outcomes, satisfaction, efficiency, and attitudes: a systematic review. *Annals of Internal Medicine* 2013;159(10):677-87.
- Otte-Trojel T. How outcomes are achieved through patient portals: a realist review. *Journal of the American Medical Informatics Association* 2014;21:751-7.
- Irizarry T, DeVito Dabbs A, Curran CR. Patient portals and patient engagement: a state of the science review. *Journal of Medical Internet Research* 2015;17(6):e148

Title:

It would be helpful for readers if the intervention under analysis was more accurately referred to in the title (as mentioned above) and the synthesis approach made explicit in the title (e.g. systematic review and narrative synthesis).

Introduction:

Beyond the overarching issues noted above, I think the introduction would benefit from further detail on the suitability and scope of the framework the authors intend to use (i.e. is the focus of the review on quality, or on both quality and safety? If so, what is the approach to safety? Is it as embedded domain in the IOM framework? Is patient safety being conceptualised as 'avoidance of potential harms' only?).

Line 100, the authors state the importance of weighing both the risks and benefits, which I agree is key. However, the risks aspect seems to get lost later on in the protocol and the review seems to then go on to focus on benefits only (e.g. aim 2, line 155).

Lines 143-145, I think it would be relevant to consider the wider context when it comes to framing the issue of resources being allocated to facilitate patient access to data, other than efficiency or

effectiveness. There is probably something to be said about other potential drivers for this, such as debates around patients' rights and data ownership in the digital era or around improving patient-centeredness of health services.

Lines 147-148, the authors could expand on why is important that aspects of quality of care other than patient-centeredness (such as timeliness or equity) are not being addressed, as this might be an interesting novel contribution of this review, particularly given that interventions aimed at improving quality of care do not necessarily encompass all the domains associated to healthcare quality, but quite often aim to improve quality of care by addressing one specific domain (e.g. patient safety).

Aims:

In my opinion, the review aims are too broadly formulated, they should enable the reader to accurately assess the suitability of the proposed methods. Again, I refer to the issue of the approach to synthesis outlined above, as the review questions are the key starting point for this.

Methods:

Search strategy:

From the major electronic databases I missed the inclusion of SCOPUS, which the authors may want to consider adding.
Are the authors considering searching any grey literature sources?
Are the authors considering the inclusion of any conference proceedings sources, or search proceedings from specific conferences for eligible studies?
Are the authors considering handsearching any key journals for eligible studies that may have been missed via title/abstract searches or controlled vocabulary?
Are the authors considering searching any trial registers to identify any ongoing (or recently completed) studies?
Are there any reasons for language restriction? If so, I suggest this could be briefly stated.

Quality assessment:

How will authors deal with studies where risk of bias is scored as high or unclear? This should be explained in the protocol. Are there any exclusion criteria associated with quality assessment? (i.e. Are the authors planning to exclude studies scored as high risk of bias on any of the five levels assessed?) Do the authors plan to conduct sensitivity analysis for any studies scored as high or unclear risk of bias? How the authors plan to report on risk of bias in the review?

Synthesis methods section (lines 215-229):

In addition to addressing the core issue noted above, regarding clarification of the synthesis approach that will be used and its appropriateness in relation to the review aims and questions, this section should also detail how the authors intend to employ the IOM framework in the context of their approach to synthesis.

I hope these suggestions are helpful to the authors.

REVIEWER	Claudia Habl Austrian Public Health Institute GOeG, Vienna, Austria
REVIEW RETURNED	07-Feb-2018

GENERAL COMMENTS	A few comments: The geographic area to be covered is not mentioned. If it is "worldwide" then the heterogeneity of both, EHR and the measured potential outcomes, briefly mentioned by the authors are not addressed good enough. They see potential bias in the heterogeneity but I miss ideas how they want to minimise this risk. Also they do not mention or consider the fact that a patient who demands/consults his/her EHR is more likely to be an "health literate" patient, so outcomes are likely to be biased. Could be used in the categorisation of studies. Still it is an interesting research question so I would recommend a go ahead. In the introduction part (row 88) authors shall add when patients have the right to access their data since 1998 (England, UK, US, EU, worldwide)? Ad PICO: Regarding interventions pls. consider to include imprints of EHR that patients in some countries (e.g., Austria) receive/d in an automated manner. reg. search terms: pls. consider the term e-Health or E-health, for instance in PubMed
---

VERSION 1 – AUTHOR RESPONSE

[Reviewer: 1 Albert Farre
Institution and Country: University of Birmingham, UK]

Comment 1) 'Thank you for providing me with the opportunity to review this manuscript, which is clearly written and well structured, addressing important and timely issues relating to patients' access to electronic health records. Overarching considerations: In my opinion, attention should be paid to accuracy and consistency in the use of terminology in relation to both their approach to synthesis and the intervention under analysis, to ensure that the proposed work is well understood.'

- Reply: We appreciate the positive feedback from the reviewer. With regards to overarching considerations, as suggested by the reviewer, we have reviewed carefully the entire manuscript and improved consistency. Specific terminology issues raised by the reviewer have been addressed as outlined in the following responses that follow, and changes made have been highlighted in the revised manuscript.

Comment 2) 'Approach to synthesis: The paper is initially framed as a mapping exercise informed by a particular framework (IOM's domains of quality of care) and as a narrative synthesis, and there is also the aim to develop a conceptual model. These would suggest that a thematic (i.e. narrative) or framework based synthesis method would be appropriate; however, in the methods section (lines 215-229) the authors outline an analysis plan relying on methods that produce descriptive and/or inferential statistics, which would imply a different approach to synthesis. This should be clarified and efforts made to ensure consistency throughout the protocol, particularly in relation to the review aims and questions.'

- Reply: For the second objective (i.e. 'we will develop a conceptual model integrating the contribution of these interventions in short-, medium- and long-term perspectives'), we aim to map the demonstrated benefits and risks in short-, medium- and long-term perspectives, by performing subgroup analysis to each one of the time frames. In this context, we acknowledge that the initial

phrasing ('to develop a conceptual model') may be misleading and wrongly suggest the need for a qualitative method (i.e. framework analysis). We have improved the terminology by replacing this expression by 'we will map the contribution of these interventions in short-, medium- and long-term time frames' (lines 48-50, 63-66).

Comment 3) 'Intervention under analysis: The review is described as exploring 'the impact of electronic data sharing on quality of care and safety' and the interventions of interest are referred to as 'EHR-based data sharing' interventions throughout. However, based on both the introduction provided and my understanding of the proposed protocol, the review is specifically concerned with interventions relating to patients' access to EHR. In my opinion this should be clarified and accurate terminology used, given that the term 'data sharing' in the context of healthcare information technologies could relate to a vast range of issues and interventions (e.g. interfacing issues between different systems within organisations, access to patient data across organisations and levels of care, etc.).'

- Reply: We thank the reviewer for highlighting this inconsistency and we agree the review concerns interventions to patient's access to EHR. The terminology used initially (EHR data-sharing interventions) may be confusing; as a result, we have replaced this wording with 'patients' access to EHR' or 'sharing EHR with patients' throughout the document (Lines 39, 98, 107, 121, 130).

Comment 4) 'Tied to the latter, if the review is specifically concerned with interventions relating to patients' access to EHR, it is paramount that the authors' address in the introduction how their proposed review differs from the following Cochrane Review protocol: Ammenwerth E, Lannig S, Hörbst A, Muller G, Schnell-Inderst P. Adult patient access to electronic health records (Protocol). Cochrane Database of Systematic Reviews 2017, Issue 6. Art. No.: CD012707. DOI: 10.1002/14651858.CD012707. Similarly, further clarification could be provided in the introduction as to what this review adds in relation to other relevant existing reviews (in addition to refs 17-20 and 24) which seem to have been missed:

- Ammenwerth E, Schnell-Inderst P, Hoerbst A. The impact of electronic patient portals on patient care: a systematic review of controlled trials. *Journal of Medical Internet Research* 2012;14(6):e162.
- Goldzweig CL, Orshansky G, Paige NM, Towfigh AA, Haggstrom DA, Miake-Lye I, et al. Electronic patient portals: evidence on health outcomes, satisfaction, efficiency, and attitudes: a systematic review. *Annals of Internal Medicine* 2013;159(10):677-87.
- Otte-Trojel T. How outcomes are achieved through patient portals: a realist review. *Journal of the American Medical Informatics Association* 2014;21:751-7.
- Irizarry T, DeVito Dabbs A, Curran CR. Patient portals and patient engagement: a state of the science review. *Journal of Medical Internet Research* 2015;17(6):e148'

- Reply: Similarly to other systematic reviews previously published, the Cochrane review protocol from Ammenwerth E et al [2017], does not systematically address all domains of quality of care; in particular, the impact of sharing EHR with patients on timeliness or equity has not been included. This is a particularly relevant gap in knowledge, given that interventions aimed at improving the quality of care do not necessarily improve all specific domains, and may even have a deleterious effect in some of them. This gap is present in the systematic reviews performed to date, including the works from Ammenwerth E [2012], Goldzweig CL [2013], Davis-Giardina T [2014] and Mold F [2015] (Lines 160-164).

Furthermore, although these reviews were unable to demonstrate clear benefits on efficiency and effectiveness measures, resources continued to be allocated to interventions and platforms aiming to share EHR with patients. As consequence of these efforts, it is plausible that studies performed in the last 5 years can provide further clarification for this evidence gap (Lines 152-159). This review will expand on the above-mentioned work, in order to comprehensively appraise the impact of sharing EHR with patients on all domains of quality of care, in order to identify recent methodological and scientific progress until June 2017.

The papers from Irizzary [2015] and Otte-Trojel [2014] do not focus specifically on the impact of sharing EHR on the quality of care; while Izzary [2015] provides a summary on how to support patient engagement through patient portals, Otte-Trojel [2014] focuses on the different mechanisms reported to yield outcome improvements. However, we thank the reviewer for identifying these studies and still incorporated their findings in the introduction. ('Patients' willingness and ability to access their health information through web portals is influenced by both individual (i.e. age, ethnicity, education level, health literacy and health status) and by health care delivery factors (i.e. provider endorsement and portal usability) [[Iziry 2015, Goldzweig CL 2013], (Lines 101-104)

Comment 5) 'Title: It would be helpful for readers if the intervention under analysis was more accurately referred to in the title (as mentioned above) and the synthesis approach made explicit in the title (e.g. systematic review and narrative synthesis).'

- Reply: The title has been edited to make the intervention under analysis and the synthesis approach more explicit. The new title reads: 'Impact of sharing electronic health records with patients on the quality and safety of care: a systematic review and narrative synthesis protocol' (Lines 1-3).

Comment 6) 'Introduction: Beyond the overarching issues noted above, I think the introduction would benefit from further detail on the suitability and scope of the framework the authors intend to use (i.e. is the focus of the review on quality, or on both quality and safety? If so, what is the approach to safety? Is it as embedded domain in the IOM framework? Is patient safety being conceptualised as 'avoidance of potential harms' only?).'

- Reply: The intended focus of the review is the impact on the six domains of health care quality, which, as the reviewer noted, include safety as an embedded domain in the IOM framework. In this work, we used the definition of patient safety proposed by the IOM and the World Health Organisation, i.e. 'prevention of errors and adverse effects to patients associated with health care'. This definition was also clarified in the manuscript (Lines 113-114).
(WHO[source]: <http://www.euro.who.int/en/health-topics/Health-systems/patient-safety>)
(IOM[source]: <https://www.ahrq.gov/professionals/quality-patient-safety/talkingquality/create/sixdomains.html>)

Comment 7) 'Line 100, the authors state the importance of weighing both the risks and benefits, which I agree is key. However, the risks aspect seems to get lost later on in the protocol and the review seems to then go on to focus on benefits only (e.g. aim 2, line 155).'

- Reply: As suggested by the reviewers, we ensured that the importance of assessing both risks and benefits is consistent and present throughout the manuscript. Accordingly, we updated Aim 2: 'To assess the demonstrated risks and benefits of these interventions (...)' and the wording throughout the manuscript (Lines 39, 63, and throughout the manuscript)

Comment 8) 'Lines 143-145, I think it would be relevant to consider the wider context when it comes to framing the issue of resources being allocated to facilitate patient access to data, other than efficiency or effectiveness. There is probably something to be said about other potential drivers for this, such as debates around patients' rights and data ownership in the digital era or around improving patient-centeredness of health services.'

- Reply: As suggested by the reviewers, we acknowledged other drivers and contributors to the increased allocation of resources to facilitate patients' access to EHR, including the arguments around patients' rights and data ownership, and the increased need to improve patient centredness in health care delivery (Line 154-158: 'Although these reviews were unable to demonstrate clear benefits on efficiency and effectiveness measures, the debates around patients' rights and data ownership in the digital era, and the need to improve patient-centredness of health care delivery have acted as strong drivers to allocate resources to interventions and platforms aiming to share EHR with patients.')

Comment 9) 'Lines 147-148, the authors could expand on why is important that aspects of quality of care other than patient-centeredness (such as timeliness or equity) are not being addressed, as this might be an interesting novel contribution of this review, particularly given that interventions aimed at improving quality of care do not necessarily encompass all the domains associated to healthcare quality, but quite often aim to improve quality of care by addressing one specific domain (e.g. patient safety).'

- Reply: To date, most reviews address a single, or a limited number of domains of quality of care. In particular, the impact of sharing EHR with patients on timeliness or equity has been sparsely addressed. This is a particularly relevant gap in knowledge, given that interventions aimed at improving quality of care do not necessarily improve all specific domains, and may even have a deleterious effect in some of them.' (adapted according to the reviewer's comment, Line 162-164).

Comment 10) 'Aims: In my opinion, the review aims are too broadly formulated, they should enable the reader to accurately assess the suitability of the proposed methods. Again, I refer to the issue of the approach to synthesis outlined above, as the review questions are the key starting point for this.'

- Reply:

The second objective was reformulated into 'we will map the contribution of these interventions in short-, medium- and long-term time frames' (Lines 48-50, 65-66, 175-176). Accordingly, the approach to synthesis will include a narrative statement supported by a summary of findings table, and subgroup analysis will be performed for each on the timeframes and domains of quality of care. Whenever possible, continuous and dichotomous outcomes will be pooled together for meta-analysis purposes. (Lines 253-254).

Comment 11) 'Methods: Search strategy:

Are the authors considering searching any grey literature sources? Are the authors considering the inclusion of any conference proceedings sources, or search proceedings from specific conferences for eligible studies? Are the authors considering searching any trial registers to identify any ongoing (or recently completed) studies? From the major electronic databases I missed the inclusion of SCOPUS, which the authors may want to consider adding. Are the authors considering hand searching any key journals for eligible studies that may have been missed via title/abstract searches or controlled vocabulary? Are there any reasons for language restriction? If so, I suggest this could be briefly stated.'

- Reply:

As suggested by the reviewer, we have added a paragraph that outlines the search strategy for grey literature, including conference proceedings: 'We will also search grey literature sources, including registrations in the International Prospective Register of Systematic Reviews (PROSPERO), reports of relevant stakeholder organisations (NHS Digital, AMIA, eHealth at WHO, International Society for Telemedicine and eHealth), and conference proceedings (last 5 years) of several related conferences (American Medical Informatics Association [AMIA], MedInfo, Medicine 2.0, Medicine X), in order to identify possible additional studies that meet the inclusion criteria.' (Line 184-190).

Regarding the reviewer's comment on the inclusion of SCOPUS, as part of our search strategy, we covered the majority of the sources of SCOPUS relevant for this work, including Pubmed, EMBASE, and conference proceedings. Of note, SCOPUS has 100% MEDLINE coverage, 100% of EMBASE coverage and 100% of Compendex coverage (engineering literature database) (Burnham J. Scopus database: a review. Biomed Digit Libr 2006).

Concerning the reviewer's comment on the identification of ongoing studies, this review aims to systematically appraise the demonstrated impact of sharing EHR with patients, (and therefore only completed studies will be included).

Regarding the potential need to hand search key journals mentioned by the reviewer, we optimised the sensitivity of our searches via controlled vocabulary with a comprehensive search strategy,

designed in collaboration with a library technician, thus minimising the need to hand search key journals.

Concerning our language restrictions, both abstract screening and full-text screening will be performed at least by two independent reviewers, and therefore they are required to be fluent in the same language. Including abstracts in other languages would, therefore, impact the feasibility of the screening process. However, this methodological choice did not result in a substantial exclusion of papers, as the proportion of papers in other languages identified in the initial screening was <4%.

Comment 12) 'Quality assessment: How the authors plan to report on risk of bias in the review? How will authors deal with studies where risk of bias is scored as high or unclear? This should be explained in the protocol. Are there any exclusion criteria associated with quality assessment? (i.e. Are the authors planning to exclude studies scored as high risk of bias on any of the five levels assessed?) Do the authors plan to conduct sensitivity analysis for any studies scored as high or unclear risk of bias?'

- Reply:

The risk of bias for each outcome across individual studies will be reported according to 1) Cochrane Risk of Bias Tool (randomized controlled trials and cluster randomized trials), 2) ROBINS-I Risk of Bias tool (quasi-experimental, cohort and case control studies) and c) the Drummond's checklist for cost-effectiveness studies (Lines 230-238). Two independent reviewers will score the selected studies and disagreements will be resolved by a third person. The information will be summarised as a narrative statement, and supported by a risk of bias table; a review-level narrative summary of the risk of bias will also be provided. (Lines 241-242).

For studies with a high or unclear risk of bias, defined as a high or unclear risk in 50% or more of the quality assessment outcomes, a narrative description of the risk of bias will be provided (Lines 243-245).

Risk of bias assessments will be incorporated into synthesis by performing sensitivity analysis (i.e. limiting to studies at lowest risk of bias in a secondary analysis)'. (Lines 247-249)

Comment 13) 'Synthesis methods section (lines 215-229): In addition to addressing the core issue noted above, regarding clarification of the synthesis approach that will be used and its appropriateness in relation to the review aims and questions, this section should also detail how the authors intend to employ the IOM framework in the context of their approach to synthesis.'

- Reply: As mentioned previously, we have improved the terminology by updating Aim 2 to 'map the contribution of these interventions in short-, medium- and long-term time frames' (lines 50-51, 65). The synthesis approach will include a narrative synthesis and a summary of findings table, supported by a subgroup analysis by time frame (Lines 246-247).

Planned subgroup analysis will also be performed by domain of quality of care (IOM framework).

Similarly, the synthesis approach will include a narrative synthesis supported by a summary of findings table for each one of the domains. (Lines 246-247). Whenever possible, continuous and dichotomous outcomes will be pooled together for meta-analysis purposes. (Lines 253-254).

[Reviewer: 2

Reviewer Name: Claudia Hahl

Institution and Country: Austrian Public Health Institute GOeG, Vienna, Austria]

Comment 1) 'The geographic area to be covered is not mentioned. If it is "worldwide" then the heterogeneity of both, EHR and the measured potential outcomes, briefly mentioned by the authors are not addressed good enough. They see potential bias in the heterogeneity but I miss ideas how they want to minimise this risk. Also they do not mention or consider the fact that a patient who demands/consults his/her EHR is more likely to be a "health literate" patient, so outcomes are likely to be biased. Could be used in the categorisation of studies. Still it is an interesting research question so I would recommend a go ahead.'

- Reply: Due to the paucity of papers published in the area, we decided not to specify a given geographic location. Although we acknowledge that this methodological choice will add an increased heterogeneity due to the diverse geographic and cultural backgrounds included (which will be discussed in the limitations of the systematic review), this strategy will optimise the literature coverage and provide a comprehensive overview of the topic.

To mitigate against the heterogeneity of outcomes, we will perform planned subgroup analysis for each of the domains of quality of care (l. 246). Furthermore, to minimise the effect of heterogeneity, parallel-group trials that are deemed comparable in relevant ways will be pooled together for a summary effect, and continuous and dichotomous outcomes will be pooled together for meta-analysis purposes (Lines 252-254). All effect sizes will be transformed into a common metric, in order to make them comparable across studies — the bias-corrected standardized difference in means (Hedges' g) — classified as positive when in favor of the intervention and negative when in favor of the control. (Lines 252-257)

We acknowledge the possibility of a health literacy-led selection bias, which could be mitigated by undertaking subgroup analysis. Unfortunately most studies do not provide a baseline assessment of participants' health literacy levels, and so this option is not feasible. This issue will be fully discussed, and critically appraised, in when discussing the limitations of our findings.

Comment 2) 'In the introduction party (row 88) authors shall add when patients have the right to access their data since 1998 (England, UK, US, EU, worldwide)?'

- Reply: This information refers to England, and was updated accordingly in the manuscript. Line 92: Although, in England, patients have had the legal right to access their health records since 1998, access to paper-based health records is mediated by health professionals and data controllers, through a cumbersome procedural process.

Comment 3) 'Ad PICO: Regarding interventions pls. consider to include imprints of EHR that patients in some countries (e.g., Austria) receive/d in an automated manner.'

- Reply: We thank the reviewer for her comment; however, we decided not to include paper-based EHR sharing interventions as these exhibit substantial differences; for instance, they do not require basic digital skills or literacy, as necessary when patient are provided with direct access to EHR. Therefore, their inclusion would increase the heterogeneity of the studies included. Instead, we have decided to comment on paper-based EHR studies in the discussion section of our systematic review.

Comment 4) 'reg. search terms: pls. consider the term e-Health or E-health, for instance in PubMed'

- Reply: The suggestion of including e-Health or E-health was tested a priori, but the corresponding MeSH term (Pubmed) for eHealth is 'Telemedicine', which breaks down to 'Remote consultation', 'Telepathology', 'Teleradiology' and 'Telerehabilitation', which was included in our final search strategy. As a result, we believe that our search strategy provides is inclusive (see below, and in Table 1).

1. (((electronic* or online or on-line or digital*) N1 (health record* or medical record* or personal record* or patient record*)) or EHR# or EMR# or ephr#)
2. ((information or data) N4 (shar* or exchang*)) or HIE or HIEs or access*)

Finally, as required by additional requirements implemented in order to include a 'Patient and Public Involvement' statement, we added a full paragraph addressing the information suggested in the introductions (Line 263-274).

REVIEWER	Claudia Habl Austrian Public Health Institute GÖG, Austria
REVIEW RETURNED	15-Apr-2018

GENERAL COMMENTS	The revised document has much improved but it would be good to give a bit a forecast of the expected results.
---

REVIEWER	Albert Farre University of Birmingham, UK
REVIEW RETURNED	23-Apr-2018

GENERAL COMMENTS	I would like to thank the authors for taking the time to address my comments. I think all the reviewers' comments have been appropriately considered and addressed by the authors in this revised version of the manuscript and look forward to reading the findings of this review in the future.
--

VERSION 2 – AUTHOR RESPONSE

- Reviewer 1: “I would like to thank the authors for taking the time to address my comments. I think all the reviewers' comments have been appropriately considered and addressed by the authors in this revised version of the manuscript and look forward to reading the findings of this review in the future.”
 - o We appreciate the positive feedback from the reviewer. His comments and useful insights were extremely valuable and greatly contributed to improve the present work.

- Reviewer 2: “The revised document has much improved but it would be good to give a bit a forecast of the expected results. “
 - o We appreciate the positive feedback from the reviewer, and thank her for the new suggestion to describe the expected results. However, in line with the journal policy for protocol articles - and in order to keep an unbiased, open approach - we decided not to incorporate it in the present version of the manuscript.

VERSION 3 – REVIEW

REVIEWER	Albert Farre University of Birmingham, UK
REVIEW RETURNED	16-Jun-2018

GENERAL COMMENTS	The revised version of the manuscript presents a complete and well written protocol. Comments previously raised have been appropriately addressed both in the manuscript and the authors' response.
---